# Comparison of Cancer Patients to Non-Cancer Patients among COVID-19 Inpatients at a National Level

**DOI:** 10.3390/cancers13061436

**Published:** 2021-03-21

**Authors:** Alain Bernard, Jonathan Cottenet, Philippe Bonniaud, Lionel Piroth, Patrick Arveux, Pascale Tubert-Bitter, Catherine Quantin

**Affiliations:** 1Department of Thoracic and Cardiovascular Surgery, CHU Dijon Bocage Hospital, 14 rue Gaffarel, BP 77908, 21079 Dijon, France; alain.bernard@chu-dijon.fr; 2Biostatistics and Bioinformatics (DIM), University Hospital, Bourgogne Franche-Comté University, BP 77908, 21079 Dijon, France; jonathan.cottenet@chu-dijon.fr; 3Faculty of Medicine, Bourgogne Franche-Comté University, 21000 Dijon, France; philippe.bonniaud@chu-dijon.fr; 4Reference Center for Rare Pulmonary Diseases, Pulmonary Medicine and Intensive Care Unit Department, University Hospital, BP 77908, 21079 Dijon, France; 5INSERM, LNC UMR1231, LipSTIC LabEx Team, 21000 Dijon, France; lionel.piroth@chu-dijon.fr; 6Inserm, CIC 1432, Clinical Investigation Center, Clinical Epidemiology/Clinical Trials Unit, Dijon University Hospital, 21000 Dijon, France; 7Infectious Diseases Department, University Hospital, BP 77908, 21079 Dijon, France; 8Center for Primary Care and Public Health, Unisanté, University of Lausanne, 1015 Lausanne, Switzerland; Patrick.Arveux@unisante.ch; 9High-Dimensional Biostatistics for Drug Safety and Genomics, Paris-Saclay University, UVSQ, Inserm, CESP, 94800 Villejuif, France; pascale.tubert@inserm.fr

**Keywords:** COVID-19, cancer, tumour subtype, mortality, intensive care unit, medico-administrative data, SARS-CoV-2, France

## Abstract

**Simple Summary:**

Several smaller studies have shown that COVID-19 patients with cancer are at a significantly higher risk of death. Our aim was to compare patients hospitalized for COVID-19 with cancer to those without cancer using national data and to study the effect of cancer on the risk of hospital death and intensive care unit admission. This study shows that, in France, patients with COVID-19 and cancer have a two-fold risk of death when compared to COVID-19 patients without cancer. This study also provides information about the types of cancer for which the prognosis is worse, such as hematological cancers and, among solid tumours, all metastatic cancers but also lung cancers. Our results reinforce the need to implement an organization within facilities to prevent the contamination of patients being treated for cancer and the importance of all measures of physical prevention and of vaccination.

**Abstract:**

(1) Background: Several smaller studies have shown that COVID-19 patients with cancer are at a significantly higher risk of death. Our objective was to compare patients hospitalized for COVID-19 with cancer to those without cancer using national data and to study the effect of cancer on the risk of hospital death and intensive care unit (ICU) admission. (2) Methods: All patients hospitalized in France for COVID-19 in March–April 2020 were included from the French national administrative database, which contains discharge summaries for all hospital admissions in France. Cancer patients were identified within this population. The effect of cancer was estimated with logistic regression, adjusting for age, sex and comorbidities. (3) Results: Among the 89,530 COVID-19 patients, we identified 6201 cancer patients (6.9%). These patients were older and were more likely to be men and to have complications (acute respiratory and kidney failure, venous thrombosis, atrial fibrillation) than those without cancer. In patients with hematological cancer, admission to ICU was significantly more frequent (24.8%) than patients without cancer (16.4%) (*p* < 0.01). Solid cancer patients without metastasis had a significantly higher mortality risk than patients without cancer (aOR = 1.4 [1.3–1.5]), and the difference was even more marked for metastatic solid cancer patients (aOR = 3.6 [3.2–4.0]). Compared to patients with colorectal cancer, patients with lung cancer, digestive cancer (excluding colorectal cancer) and hematological cancer had a higher mortality risk (aOR = 2.0 [1.6–2.6], 1.6 [1.3–2.1] and 1.4 [1.1–1.8], respectively). (4) Conclusions: This study shows that, in France, patients with COVID-19 and cancer have a two-fold risk of death when compared to COVID-19 patients without cancer. We suggest the need to reorganize facilities to prevent the contamination of patients being treated for cancer, similar to what is already being done in some countries.

## 1. Introduction

Coronavirus Disease (COVID-19) is a viral infectious disease caused by the new SARS-Cov-2 coronavirus. Symptoms typically start 2 to 14 days after exposure and are characterized by general clinical signs and taste and smell disorders. COVID-19 can be complicated by pneumopathy, acute respiratory distress syndrome and lead to death [1,2,3]. Other serious presentations include cardiovascular or digestive problems.

As of January 28, 2021, the COVID-19 epidemic was responsible for 99,727,853 cases confirmed worldwide since Dec 31, 2019, including 32,218,360 in Europe, and 2,137,670 deaths worldwide, of which 166,613 deaths were in Europe (source: ECDC). In France, there were 3,053,617 confirmed cases and 73,049 deaths [4,5]. Based on expert opinion and on data from the literature [6,7,8,9,10], the high council of public health (Haut Conseil de la Santé Publique, HCSP) considers that individuals most at risk of developing a serious form of SARS-Cov-2 infection are people aged 70 years and older, patients with cardiovascular conditions (complicated hypertension, stroke or coronary heart disease, heart failure), insulin-dependent unbalanced diabetics or presenting secondary complications, and patients with chronic respiratory disease, chronic kidney failure dialysis or treatment for cancer. Another group considered at risk [11,12,13,14] is patients with congenital or acquired immunosuppression (cancer chemotherapy, immunosuppressive therapy, biotherapy and/or corticosteroids with immunosuppressive dose, uncontrolled HIV infection or with CD4 < 200/mm^3^, those who have had a solid organ or hematopoietic stem cell transplant linked to a malignant hematology during treatment), and individuals with morbid obesity.

Patients with cancer are considered immunosuppressed and are at a higher risk of infection. This immunosuppression is caused by the disease and specific treatments such as chemotherapy or surgery. These hypotheses have been confirmed by publications [7,8,9,13,15] reporting that patients with cancer are three times more likely to develop serious complications from COVID-19 than those without cancer. Wang et al. [13] addressed the severity of complications in patients with malignancy. In the report by Dai et al. [16], patients with cancer were significantly more likely to require ICU admission, have severe/critical symptoms or die, after adjustment for covariates. Deng et al. [17] also found a significantly higher risk of death for patients with cancer than those without. In contrast, Brar et al. [18] observed that patients with COVID-19 and cancer had similar outcomes compared with matched patients without cancer in two NYC hospitals. Most of these studies are of small size, and their results are not fully applicable to the French population, due to differences in population structure, but also different genetic, behavioral and cultural factors.

Using data from the French national hospital database, which includes the data collected for close to 100,000 hospitalized COVID-19 patients over a two-month period, our goal is to compare patients hospitalized for COVID-19 with cancer (with or without metastasis) to those without cancer, and to study the effect of cancer on the risk of hospital death and ICU admission.

## 2. Methods

### 2.1. Database

A retrospective cohort study was conducted using the national *Programme de Médicalisation des Systèmes d’Information* (PMSI) database, which is designed to include discharge summaries for all inpatient admissions to public and private hospitals in France. Inspired by the American DRG (diagnosis-related groups) model, the information in these abstracts is anonymous and covers both medical and administrative data. Diagnoses identified during the hospital stay are coded according to the 10th edition of the International Classification of Diseases (ICD-10), and procedures performed during hospitalization are coded according to the French Common Classification of Medical Procedures.

In France, Stage 1 of the COVID-19 epidemic was declared on 23 February, Stage 2 on 29 February, and Stage 3 on 14 March 2020. The first peak of the epidemic occurred during the second week of April 2020. During stages 1 and 2, all patients with COVID-19 had to be hospitalized regardless of clinical presentation, while only those requiring hospital care for their clinical condition were admitted to hospital during Stage 3. A PMSI database was developed for COVID-19, and hospitals were asked to perform accelerated data transmission for COVID-19 patients from March 2020 onwards at the request of the government, according to the decree dated 21 April 2020.

### 2.2. Population

All patients hospitalized for or with COVID-19 from 1 March to 30 April 2020, were included, regardless of age. Patients were followed up until the end of their hospital stay, even if that date was after April. Hospital stays for COVID-19 were identified by primary diagnoses (PD), related diagnoses (RD) or associated diagnoses (AD) by International Classification of Diseases, 10th revision (ICD-10) codes U0710, U0711, U0712, U0714 or U0715. We identified cancer patients using ICD-10 codes in PD, RD or AD, corresponding to malignant tumors (all those beginning with ‘C’), with and without metastasis, and separated them into different tumor subtypes (breast, colorectal, prostate, lung, digestive non-colorectal, urinary tract, female genital organs, lip-oral cavity-pharynx, skin, hematological, other cancers including mesothelial and soft issue, respiratory and intrathoracic organs (except lung), bone and articular cartilage, endocrine glands, other male cancers and central nervous system). ICD-10 codes used are presented in Appendix A.

This study was approved by the *Comité Ethique et Scientifique pour les recherches, les études et les évaluations dans le domaine de la santé* (CESREES, Ethics and Scientific Committee for Research, Studies and Evaluation in Health, 9 June 2020) and the *Institut des Données de Santé* (INDS, French Institute of Health Data, registration number 1611357, 15 June 2020) and authorized by the *Commission Nationale de l’Informatique et des Libertés* (CNIL, French Data Protection Authority, registration number DR-2020-250, 3 July 2020).

### 2.3. Outcomes

Our primary outcome was transfer to intensive care unit (ICU), determined by the presence of an ICU stay indicator in the filed claims. Our second outcome was in-hospital death, defined as any patient who died in hospital during their COVID-19 stay.

### 2.4. Variables

The following variables were also extracted for each inpatient stay: age (nine age classes were defined: <18, 18–30, 31–40, 41–50, 51–60, 61–70, 71–80, 81–90 and >90), sex, and all diagnoses (PD, RD and AD) recorded in the discharge abstracts in order to analyze comorbidities (hypertension, diabetes, dementia, HIV, heart failure, chronic respiratory and kidney diseases, cirrhosis, peripheral vascular disease, overweight, dyslipidaemia, deficiency anemia, pulmonary bacterial infection) and complications (acute respiratory and kidney diseases, stroke, myocardial infarction, atrial fibrillation, venous thrombosis including pulmonary embolism). If any condition was listed on the discharge abstract during a stay for COVID-19, then the patient was considered to have had the condition. Hospital stays were also characterized according to the type of hospital, private or public. We also used the social deprivation index in order to take into account the patient’s socio-economic environment [19]. This index ranges theoretically from −6.4 (the least deprived areas) to 21.2 (the most deprived areas). We divided this index into 4 quartiles from the overall population, with the lowest quartile representing the least deprived.

### 2.5. Statistical Analysis

Qualitative variables are provided as frequencies (percentages), and quantitative variables are provided as means ± standard deviation (SD) and medians (interquartile range (Q1–Q3)).

Qualitative variables were compared using the Chi-2 test or the Fisher’s exact test, and the Student’s t test or Mann–Whitney test were used to compare quantitative variables.

Our outcomes were first compared for all patients, then separately for metastatic cancer, and then by age group and tumor subtypes (in particular, we differentiated between solid cancers and hematological cancers). The results were given for patients with only one tumor subtype for this sub-analysis (patients with 2 tumor subtypes were excluded).

We used logistic regression models to estimate the effect of cancer on the risk of in-hospital death and transfer to ICU, adjusting for age, sex and other comorbidities after studying correlations. All models were constructed using backward selection.

Mortality rates for each tumor subtype were also analyzed, and we studied the risk of hospital death among cancer patients, using colorectal cancers as the reference group, and adjusting for age, sex and other comorbidities. We used colorectal cancer as the reference for several reasons: it is one of the most common types of cancer, is almost equally distributed between men and women and can be detected easily with a free screening from the age of 50.

We also performed sensitivity analyses using 2-level hierarchical models. The first model used the individual variables as the 1st level and the type of hospital (public or private) as the 2nd level. The second model used the individual variables as the 1st level, and the social deprivation score related to geographical unit as the 2nd level.

Considering the potential heterogeneity of patients admitted to hospital in the three stages of the epidemic, we finally performed a second series of sensitivity analyses restricted to Stage 3 of the epidemic (14 March onwards).

The statistical significance threshold was set at <0.05. All analyses were performed using SAS (SAS Institute Inc., Version 9.4, Cary, NC, USA).

## 3. Results

### 3.1. Patient Characteristics

We included 89,530 patients diagnosed with COVID-19, among whom we identified 6201 cancer patients (6.9%). Only 23 patients with cancer were identified among 1227 patients aged less than 18 years with COVID-19 (less than 2%). Hematological, lung, digestive, prostate, breast and colorectal cancer were the most frequent tumor subtypes (Table 1). The main baseline characteristics of our study population are presented in Appendix A. Patients hospitalized for COVID-19 who had a cancer diagnosis were more likely to be male and older (mean age 72 vs. 65 years). Almost 50% of cancer patients ≤ 30 years old had a hematological cancer (Appendix A).

The distribution of the social deprivation score was different between the two groups, with the lowest quartile (least deprived) being more frequently found among cancer patients. We found that public hospitals admitted 89.1% of patients without cancer, 92.7% of patients with hematological cancer, 84.1% of patients with solid metastatic cancer and 88.3% of patients with solid cancer without metastasis (Table 2).

The comparison of comorbidities in patients with and without cancer is reported in Table 2 and Appendix A. Whatever the metastatic status, when compared to COVID-19 patients without cancer, patients with a solid cancer were more likely to have cirrhosis and COPD and were less often obese or overweight. Other comorbidities were distributed differently depending on the metastasis, in particular hypertension, diabetes and chronic kidney disease.

Compared to patients without cancer, patients with a hematological cancer were more likely to have hypertension, heart failure, chronic kidney disease, dyslipidaemia, deficiency anemia or pulmonary bacterial infection, and they were less likely to be obese or overweight.

The comparison of complications in patients with and without cancer is reported in Table 2 and Appendix A. Solid cancer patients with and without metastases have significantly more complications than patients without cancer, including acute respiratory failure, pulmonary embolism and venous thrombosis.

Hematological cancer patients also had significantly more complications than patients without cancer, including acute respiratory failure, venous thrombosis, septic shock, atrial fibrillation and acute kidney failure.

### 3.2. Admission to ICU

Admission to ICU was significantly less frequent (*p* < 0.01, Appendix A) among patients with cancer (15.0%) than among patients without cancer (16.4%), and the rate was even lower among metastatic solid cancer patients (8.9%, Table 2).

Regarding age, up to the age of 40, the rate of ICU admission was significantly higher among cancer patients (vs. without cancer), but significantly lower from 61 to 80 years of age (Appendix A). For metastatic solid cancer patients (compared to patients without cancer), the rate was significantly lower from 51 to 90 years (Figure 1A). After we adjusted for sex and comorbidities, metastatic solid cancer patients 50 to 90 years old were still significantly less often admitted to ICU (Table 3).

Concerning hematological cancer, admission to ICU was significantly more frequent (*p* < 0.01) among hematological cancer patients (24.8%, Table 2) than among patients without cancer (16.4%). Conversely to solid cancer, the rate of admission to ICU was significantly higher among hematological cancer patients up to 80 years (Figure 1A). After adjustment on sex and comorbidities, hematological cancer patients were still significantly more often admitted to ICU, up to 80 years (Table 3).

Appendix A shows the different admission rates according to tumor subtype and metastatic status.

### 3.3. In-Hospital Mortality

Hospital mortality rates were significantly higher (*p* < 0.01, Appendix A) among patients with cancer (33%) than among patients without cancer (15.7%), and this rate was even higher among metastatic solid cancer patients (39.0%, Table 2). The maximum time to in-hospital death observed in our study was 61 days for hematological cancers, 81 days for metastatic solid cancers, 72 for non-metastatic solid cancers and 95 days for patients without cancer.

The hospital mortality rate was significantly higher among cancer patients in all age groups, although the gap narrows in the oldest patients (Appendix A). For metastatic solid cancer patients (compared to patients without cancer), the hospital mortality rate was significantly higher from 18 to 90 years (Figure 1B).

After adjusting for age, sex and comorbidities, it was shown that solid cancer patients (compared to patients without cancer) had a significantly higher hospital mortality rate, and this difference was more marked for metastatic solid cancer patients (aOR 3.6 (95% CI 3.2–4.0), Table 4) than for patients without metastasis (aOR 1.4 (95% CI 1.3–1.5)).

Concerning hematological cancer, hospital mortality rates were significantly higher (*p* < 0.01) among hematological cancer patients (33.8%, Table 2) than among patients without cancer (15.7%). The hospital mortality rate for hematological cancer patients was significantly higher before 30 years and from 40 to 90 years (Figure 1B). After adjusting for age, sex and comorbidities, hospital mortality rates were significantly higher in hematological cancer patients than in patients without cancer (aOR = 2.2 [2.0–2.5], Table 4). These results were the same for sensitivity analyses using hierarchical models by taking into account either the hospital type or the social deprivation score (Appendix A).

Finally, Appendix A shows the hospital mortality rates according to cancer subtype and metastatic status. After adjusting for age, sex and comorbidities (Table 5), patients with lung cancer, digestive cancer (excluding colorectal cancer) and hematologic cancer had a higher hospital mortality risk (respectively aOR of 2.0 (95% CI 1.6–2.6), 1.6 (95%CI 1.3–2.1) and 1.4 (95%CI 1.1–1.8)), using colorectal cancers as the reference group. Whatever the metastatic status, this difference was still observed for digestive and lung cancers (Table 5).

These results were the same for sensitivity analyses using hierarchical models either by taking into account the hospital type or the social deprivation score (Appendix A).

Our second series of sensitivity analyses, which was restricted to Stage 3 of the epidemic, which represents 95% of the cohort (14 March onwards), provided similar results (Appendix A).

## 4. Discussion

Our results show that among patients hospitalized in France for COVID-19, those with cancer, whether metastatic or not, were more likely to have comorbidities such as cirrhosis and COPD. However, they were less often obese or overweight, and they were also less deprived than COVID-19 patients without cancer. These findings add to the existing body of literature, seeing as, so far, few studies have described the comorbidities of cancer patients diagnosed with COVID-19. In another study that included all types of cancer, Kuderer et al. [20] observed a higher prevalence of obesity (19%), but their patients were residing in North America or Spain and may therefore be difficult to compare to the French population studied here. In addition, our population is larger, seeing as it includes children and is restricted to more serious forms of COVID-19, since our entire population was hospitalized. As regards social deprivation, cancer is more frequent in low-income populations in France, with the exception of breast cancer [21]. COVID-19 also affects the most underprivileged. However, our data indicate that COVID-19 patients without cancer were more underprivileged than COVID-19 patients with cancer. Further studies are needed to better understand the relationship between COVID-19, cancer and deprivation.

Regarding the outcomes, patients with cancer had significantly more complications than patients without cancer, including acute respiratory failure, pulmonary embolism, atrial fibrillation and acute kidney failure. The hospital mortality rate was two times higher among patients with cancer than among patients without cancer, which was also observed in a large US multicentre cohort study [22], with an odds ratio for death at 28 days of 2.15. This difference was not found to be significant by Brar et al. [18], but their case-control study included only 117 patients with COVID-19 and active malignancy and 468 controls without cancer. Interestingly, in our study, this gap was larger for middle-aged patients, even though the mortality rate increased significantly with age. This observation is consistent with what was shown by Lee et al. in an English cohort of adult COVID-19 patients with cancer [23]. However, these younger patients were most often diagnosed with hematological cancers, which have a markedly high mortality rate.

Among patients with solid tumors, those with metastases also had a higher mortality curve than those without, as expected. Patients with advanced neoplastic disease are considered immunocompromised and were also found to have a very high risk of death in other studies [24,25,26,27]. This could be explained by the accumulation of risk factors related to severe COVID-19 infection and from the cancer itself or its treatment.

The mechanisms underlying the progression of COVID-19 to a severe form may not all be the same, since not all patients with cancer are at a similarly high risk of developing severe COVID-19. These mechanisms certainly include not only host-related factors such as age or cachexia, cancer- or drug-induced hypercoagulability states, or the possible hyperexpression of entry factors such as angiotensin-converting enzyme 2 or neurophilin-1 [24], but also cancer-related myeloid cell dysfunction and T-cell exhaustion, both of which impair antiviral immunity [25]. On the other hand, the immune system is also involved in the subsequent virus-driven cytokine storm [26]. Therefore, cancer- or drug-associated immune depression may temper or delay its development [27].

Some cancer types, such as digestive (excluding colorectal), lung and hematological cancer, presented a high risk of death, which is consistent with previous findings [20,23,28]. For instance, patients with lung cancer were twice as likely to die as patients with colorectal cancer. We have shown that patients with hematological cancers had very different outcomes than patients with solid tumors, and they were more likely to die than patients with solid cancer without metastasis. This finding was also reported by Lee et al. [23], but the results were not significant, possibly due to a lack of power. Other studies [23,28] have shown that among hematological cancers, acute myeloid leukemia or lymphomas led to a higher risk of death than other clinical forms. The high level of ICU access for hematological cancers also suggests more severe COVID-19 symptoms, which could be due to the treatment of these patients but also to the immunodepression that they experience during treatment and probably throughout their lives after the cessation of curative treatments [24,25,26,27].

Another novel aspect of our work is indeed the analysis of access to intensive care for patients with cancer and COVID-19. Admission to ICU was slightly less frequent for patients with cancer than for patients without cancer, but it varied according to the tumor subtype. For instance, a quarter of hematological cancer patients was admitted to ICU, while patients with solid tumours were less often admitted, regardless of metastatic status. More generally, it must be noted that patients with hematological cancer were admitted to ICU more often than other patients with COVID-19, up until the age of 80 years. One probable reason for which COVID-19 cancer patients under 40 years of age were dramatically more frequently admitted to the ICU than other COVID-19 patients is related to the age of patients with hematological cancer, who tend to be younger than patients with other types of cancer. Our study also shows that patients with solid non-metastatic cancer were more often admitted to the ICU than patients in the same age group (<40 years) without cancer. A second explanation may be that these patients are more likely to have more severe forms of COVID-19 infection. However, even if the national database made it possible to refine the analysis in terms of age classes, due to the smaller group sizes, the confidence intervals of adjusted odds ratios are large for patients younger than 40 years. On the other hand, cancer patients in the oldest age group (>80 years) had a slightly lower percentage of ICU support, although non-significant, which contrasts with their high in-hospital mortality. This could reflect the fact that patients may have been assessed for ICU access in terms of clinical status and prognosis. Overall, whether or not a patient was admitted to an ICU seemed to depend, in addition to age, on the seriousness of the COVID-19 symptoms, which may have increased the likelihood of transfer to the ICU, and on the severity/prognosis of the underlying neoplastic disease, which may have contraindicated admission to the ICU.

### 4.1. Strengths

Our work is unique for the large number of patients included and the comprehensiveness of the data. In France, the national hospital database includes information from all private and public French hospitals that treat COVID-19 and cancer patients. The fact that these national data are used for the allocation of hospital budgets encourages high levels of data coherence, accuracy and exhaustiveness. Consequently, this study includes nationwide data for nearly 90,000 hospitalized patients (89,530) with COVID-19, among whom we identified 6201 cancer patients. This study is one of the few providing data on all ages and with complete hospital follow-up, thus including all in-hospital deaths, whatever the length of stay. Moreover, we were able to obtain information on metastatic status for all solid tumours, patient co-morbidities, and a measure of social deprivation. In addition, our data set was complete. While information is sometimes not completed in the discharge abstract when there is no impact on patient care, which is often the case for outpatient care, it was not the case for the considered hospitalisations.

Finally, we conducted several sensitivity analyses that allowed us to consolidate our results. In particular, we restricted the period to stage 3 of the epidemic, showing similar results to those of the main analysis.

### 4.2. Limitations

We recognize that there are several limitations. First, we included only hospitalized patients. Moreover, we acknowledge that, despite the collection of nationwide data, the conclusions about cancer in children need to be taken with caution considering the small number of hospitalized cases and the small number of in-hospital deaths and ICU support. Secondly, there was a potential for the misclassification of comorbidities, since they were only identified during the stays for COVID-19, but this misclassification bias is likely to be non-differential for the majority of the comorbidities. Acute and chronic conditions could not always be distinguished, for example, where the same code was used for chronic heart failure and for cardiac decompensation. Another limitation of this study relates to the stage of tumor progression, seeing as we only had information on the metastatic endpoint. We had no information on the ongoing treatment at the time of the COVID-19 episode, which would have helped us to assess potentially induced immunosuppression. For instance, a patient undergoing chemotherapy may have decreased immunity with a higher risk of infection. On the contrary, it cannot be excluded that the drugs used in some conditions (e.g., checkpoint inhibitors or tyrosine kinase inhibitors) might have a therapeutic activity against SARS-CoV2, as suggested by some preliminary in vitro or clinical data. More generally, we had no data regarding treatments, so the impact of other drugs that could influence the severity of COVID-19 in hospitalized patients (e.g., dexamethasone, IL6 inhibitors, or ARA2 inhibitors) was also not assessed.

Overall, our study confirms that COVID-19 patients with cancer have a high risk of in-hospital death. To our knowledge, this is the largest study in the world so far, and the only one to provide comprehensive data for an entire country. Our study compares cancer patients with COVID-19 to non-cancer patients with COVID-19. This answers a question that was highlighted by Kuderer et al. in their pioneering work [20] as a fundamental issue for clinical management and the organization of preventive measures and that is not addressed on a large scale elsewhere. In light of our results, the measures suggested by some authors [29] to better protect cancer patients seem to be appropriate. These measures include informing doctors about the specific risks of COVID-19 for cancer patients and using an algorithm that has been developed to screen for COVID-19 in patients undergoing cancer treatment [29].

## 5. Conclusions

This study shows that, in France, patients with COVID-19 and cancer have an increased risk of mortality (two-fold risk) compared to COVID-19 patients without cancer, particularly for young patients and for patients with solid metastatic cancers, and were less often admitted in ICUs. Patients with cancer were more affected with some comorbidities than other COVID-19 patients, such as heart failure, chronic respiratory disease, peripheral vascular disease and chronic kidney disease. This study provides information about the types of cancer for which the prognosis is worse, such as hematological cancers and, among solid tumours, all metastatic cancers but also lung cancers. Our results reinforce the need to implement an organization within facilities to prevent the contamination of patients being treated for cancer and the importance of all measures of physical prevention and of vaccination.

## Figures and Tables

**Figure 1 cancers-13-01436-f001:**
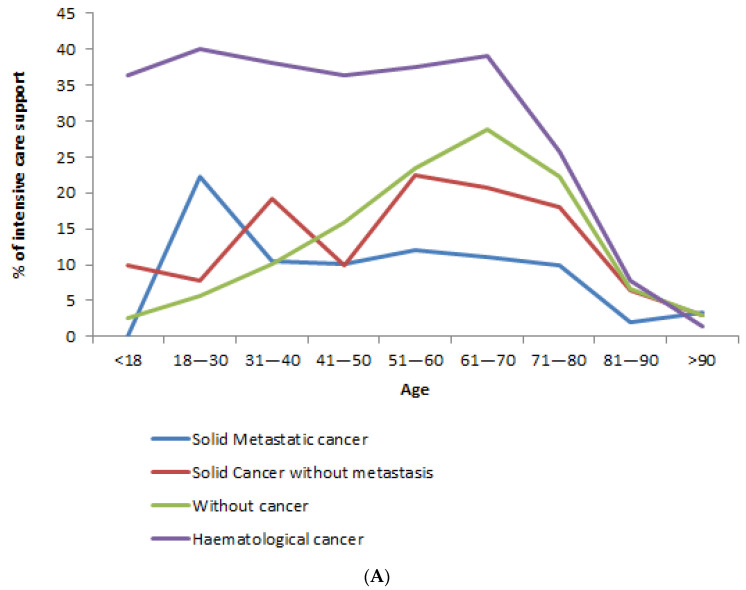
(**A**). Intensive care support rates of patients hospitalized in France for COVID-19 (from 1 March 2020 to 30 April 2020) for cancer (with or without metastasis) or not, according to their age at admission. (**B**). Hospital mortality rates of patients hospitalized in France for COVID-19 (from 1 March 2020 to 30 April 2020) for cancer (with or without metastasis) or not, according to their age at admission.

**Table 1 cancers-13-01436-t001:** Tumour subtypes for cancer patients (with or without metastasis) hospitalized in France for COVID-19 (from 1 March 2020, to 30 April 2020).

	All Cancer n = 5722		Solid Metastatic Cancer n = 1775		Solid Cancer without Metastasis n = 2558	
	*n*	%	*n*	%	*n*	%
Hematological	1.89	24.3	-	-	-	-
Lung	873	15.3	461	26.0	412	16.1
Digestive (non-colorectal)	626	10.9	244	13.8	382	14.9
Prostate	621	10.9	196	11.0	425	16.6
Breast	561	9.8	241	13.6	320	12.5
Colorectal	518	9.1	244	13.8	274	10.7
Urinary tract	363	6.3	128	7.2	235	9.2
Other cancers *	303	5.3	54	3.0	249	9.7
Female genital organs	185	3.2	119	6.7	66	2.6
Lip, oral cavity and pharynx	162	2.8	45	2.5	117	4.6
Skin	121	2.1	43	2.4	78	3.1

* including mesothelial and soft issue, respiratory and intrathoracic organs (except lung), bone and articular cartilage, endocrine glands, other cancers in males and central nervous system.

**Table 2 cancers-13-01436-t002:** Main baseline characteristics of patients hospitalized in France for COVID-19 (from 1 March 2020 to 30 April 2020) according to the presence of cancer and cancer types.

	Hematological Cancer	Solid Metastatic Cancer	Solid Cancer without Metastasis	Without Cancer	*p*-Value *
Number of patients	1389	1775	2558	83,329	
Male gender (*n*, %)	797 (57.4) **	998 (56.2) **	1627 (63.6) **	43,787 (52.6)	<0.01
Age					<0.01
Mean +/− std	72 +/− 15 **	70 +/− 13 **	74 +/− 13 **	65 +/− 20	
Med (Q1–Q3)	74 (65–83)	71 (62–80)	75 (66–84)	67 (51–81)	
Social deprivation score					
Mean +/− std	−0.44 +/− 1.81 ^$^	−0.45 +/− 1.72 ^$^	−0.44 +/− 1.79 ^$^	−0.26 +/− 1.78	<0.01
Med [Q1–Q3]	−0.17 [−1.41–0.82]	−0.20 [−1.40–0.75] ^$^	−0.20 [−1.38–0.78] ^$^	−0.14 [−1.22–0.93]	
Lowest (<−1.233)	383 (27.3) **	504 (28.1) **	702 (26.9) **	19,661 (24.8)	<0.01
Second ([−1.23; −0.146])	332 (23.7)	443 (24.7)	672 (25.7)	19,621 (24.8)	
Third ([−0.146; −0.917])	369 (26.3)	463 (25.8)	658 (25.2)	19,983 (25.2)	
Highest (≥0.917)	319 (22.7) ^$^	382 (21.3) ^$^	582 (22.3) ^$^	19,980 (25.2)	
Hospital type admission					
Public	1288 (92.7) **	1492 (84.1) ^$^	2259 (88.3)	74,248 (89.1)	<0.01
Private	101 (7.3) ^$^	283 (15.9) **	299 (11.7)	9081 (10.9)	
Comorbidities					
Hypertension	514 (37.0) **	502 (28.3) ^$^	1016 (39.7) **	27,406 (32.9)	<0.01
Dementia	81 (5.8) ^$^	59 (3.3) ^$^	213 (8.3)	6361 (7.6)	<0.01
HIV	9 (0.7)	6 (0.3)	16 (0.6)	400 (0.5)	0.39
Heart failure	161 (11.6) **	128 (7.2)	246 (9.6) **	6553 (7.9)	<0.01
Chronic respiratory disease	22 (1.6)	37 (2.1)	49 (1.9)	1313 (1.6)	0.10
Chronic kidney disease	161 (11.6) **	103 (5.8) ^$^	303 (11.9) **	6838 (8.2)	<0.01
Cirrhosis	10 (0.7)	23 (1.3) **	90 (3.5) **	584 (0.7)	<0.01
Diabetes	237 (17.1)	301 (17.0) ^$^	587 (23.0) **	15,841 (19.0)	< 0.01
Peripheral vascular disease	47 (3.4)	67 (3.8)	159 (6.2) **	2572 (3.1)	<0.01
Obese or overweight	129 (9.3) ^$^	73 (4.1) ^$^	205 (8.0) ^$^	9691 (11.6)	<0.01
Obese	104 (7.5) ^$^	66 (3.7) ^$^	170 (6.7) ^$^	8257 (9.9)	<0.01
Dyslipidemia	96 (6.9) **	78 (4.4)	173 (6.8) **	4103 (4.9)	<0.01
Deficiency Anemia	82 (5.9) **	73 (4.1)	159 (6.2) **	3162 (3.8)	<0.01
COPD	68 (4.9)	116 (6.5) **	250 (9.8) **	4385 (5.3)	<0.01
Pulmonary bacterial infection	137 (9.9) **	98 (5.5) ^$^	177 (6.9)	5845 (7.0)	0.05
Complications					
Acute respiratory failure	476 (34.3) **	519 (29.2) **	753 (29.4) **	22,436 (26.9)	<0.01
Pulmonary embolism	59 (4.3)	86 (4.9) **	105 (4.1) **	2813 (3.4)	<0.01
Venous thrombosis	89 (6.5) **	115 (6.5) **	144 (5.6) **	3988 (4.8)	<0.01
Septic shock	59 (5.0) **	26 (1.5) ^$^	86 (3.3)	2356 (2.8)	<0.01
Myocardial infarction	10 (0.7)	2 (0.1) ^$^	11 (0.4)	531 (0.6)	0.01
Atrial fibrillation	239 (17.2) **	217 (12.2)	440 (17.2) **	10,155 (12.2)	<0.01
Stroke	12 (0.9)	15 (0.9)	37 (1.5)	997 (1.2)	0.20
Hemorrhagic stroke	5 (0.4)	6 (0.3)	6 (0.2)	234 (0.3)	0.82
Ischemic stroke	5 (0.4)	6 (0.3) ^$^	24 (0.9)	674 (0.8)	0.07
Transient Ischemic Attack	2 (0.1)	5 (0.3)	8 (0.3)	146 (0.2)	0.17
Acute kidney failure	157 (11.3) **	107 (6.0)	202 (7.9) *	5258 (6.3)	<0.01
Intensive care support	345 (24.8) **	157 (8.9) ^$^	373 (14.6) ^$^	13,655 (16.4)	<0.01
In-hospital death	470 (33.8) **	693 (39.0) **	690 (27.0) **	13,057 (15.7)	<0.01

* *p*-value related to the comparison of the three groups (solid metastatic cancer, solid cancer without metastasis and without cancer). ** significantly higher in the cancer group compared to the non-cancer group (*p* < 0.05). ^$^ significantly lower in the cancer group compared to the non-cancer group (*p* < 0.05).

**Table 3 cancers-13-01436-t003:** Logistic regression to study the risk of transfer to intensive care unit (adjusted odds ratio) regarding patients hospitalized in France for COVID-19 (from 1 March 2020, to 30 April 2020) with cancer or not.

	<40	41–50	51–80	81–90	>90
Cancer	3.6 [2.4–5.6]	1.2 [0.8–1.7]	0.7 [0.6–0.8]	0.8 [0.6–0.9]	0.8 [0.4–1.7]
Without cancer	ref	ref	ref	ref	ref
Solid Cancer with metastasis	1.5 [0.6–4.0]	0.7 [0.3–1.5]	0.4 [0.3–0.5]	0.2 [0.1–0.5]	1.2 [0.3–4.9]
Solid Cancer without metastasis	2.2 [0.9–5.7]	0.6 [0.2–1.4]	0.7 [0.6–0.8]	0.9 [0.7–1.3]	1.0 [0.4–2.3]
Hematological cancer	10.4 [5.5–19.9]	3.7 [2.0–6.7]	1.5 [1.3–1.8]	1.0 [0.7–1.5]	0.5 [0.1–3.4]
Without cancer	ref	ref	ref	ref	ref

Adjusted odds ratio on sex, dementia, heart failure, chronic respiratory disease, cirrhosis, diabetes, deficiency anemia and pulmonary bacterial infection.

**Table 4 cancers-13-01436-t004:** Logistic regression to study the risk of in-hospital death regarding patients hospitalized in France for COVID-19 (from 1 March 2020, to 30 April 2020) with cancer or not.

	Death	Hospital Mortality Rate	OR	*p*-Value	aOR *	*p*-Value
Cancer	2047	33.0	2.7 [2.5–2.8]	<0.01	2.2 [2.0–2.3]	<0.01
Withour cancer	13,057	15.7	ref		ref	
Solid cancer with metastasis	693	39.0	3.4 [3.1–3.8]	<0.01	3.6 [3.2–4.0]	<0.01
Solid cancer without metastasis	690	27.0	2.0 [1.8–2.2]	<0.01	1.4 [1.3–1.5]	<0.01
Hematological cancer	470	33.8	2.8 [2.5–3.1]	<0.01	2.2 [2.0–2.5]	<0.01
Without cancer	13,057	15.7	ref		ref	

OR: odds ratio; aOR: adjusted odds ratio. * Adjusted odds ratio on sex, dementia, heart failure, chronic respiratory disease, cirrhosis, diabetes, deficiency anemia and pulmonary bacterial infection.

**Table 5 cancers-13-01436-t005:** Logistic regression to study the risk of hospital death among cancer patients hospitalized in France for COVID-19 (from 1 March 2020 to 30 April 2020), using colorectal cancers as the reference group.

	*n*	Death	Hospital Mortality Rate	OR	*p*-Value	aOR *	*p*-Value
All Cancer							
Colorectal	518	142	27.4	ref	–	ref	–
Digestive (non-colorectal)	626	233	37.2	1.6 [1.2–2.0]	<0.01	1.6 [1.3–2.1]	<0.01
Breast	561	133	23.7	0.8 [0.6–1.1]	0.16	1.0 [0.8–1.4]	0.76
Prostate	621	188	30.3	1.2 [0.9–1.5]	0.29	0.9 [0.7–1.2]	0.36
Lung	873	359	41.1	1.8 [1.5–2.3]	<0.01	2.0 [1.6–2.6]	<0.01
Urinary tract	363	122	33.6	1.3 [1.0–1.8]	0.049	1.2 [0.9–1.7]	0.15
Female genital organs	185	54	29.2	1.1 [0.8–1.6]	0.64	1.4 [0.9–2.1]	0.07
Lip, oral cavity and pharynx	162	38	23.5	0.8 [0.5–1.2]	0.32	0.9 [0.6–1.4]	0.73
Skin	121	32	26.5	1.0 [0.6–1.5]	0.83	0.8 [0.5–1.3]	0.44
Other cancers **	303	82	27.1	1.0 [0.7–1.3]	0.91	1.2 [0.8–1.6]	0.37
Hematological	1389	470	33.8	1.4 [1.1–1.7]	0.01	1.4 [1.1–1.8]	<0.01
Solid Cancer with metastasis							
Colorectal	244	84	34.4	ref	–	ref	–
Digestive (non-colorectal)	244	109	44.7	1.5 [1.1–2.2]	0.02	1.6 [1.1–2.3]	0.02
Breast	241	77	32.0	0.9 [0.6–1.3]	0.56	1.1 [0.8–1.7]	0.52
Prostate	196	74	37.8	1.2 [0.8–1.7]	0.47	0.9 [0.6–1.3]	0.73
Lung	461	215	46.6	1.7 [1.2–2.3]	<0.01	1.7 [1.2–2.4]	<0.01
Urinary tract	128	54	42.2	1.4 [0.9–2.2]	0.14	1.4 [0.9–2.1]	0.15
Female genital organs	119	41	34.5	1.0 [0.6–1.6]	0.99	1.2 [0.7–2.0]	0.44
Lip, oral cavity and pharynx	45	12	26.7	0.7 [0.3–1.4]	0.31	0.7 [0.4–1.5]	0.37
Skin	43	12	27.9	0.7 [0.4–1.5]	0.41	0.7 [0.4–1.5]	0.41
Other cancers **	54	15	27.8	0.7 [0.4–1.4]	0.35	0.8 [0.4–1.6]	0.53
Solid Cancer without metastasis							
Colorectal	274	58	21.2	ref	–	ref	–
Digestive (non-colorectal)	382	124	32.5	1.8 [1.2–2.6]	<0.01	2.0 [1.3–3.0]	<0.01
Breast	320	56	17.5	0.8 [0.5–1.2]	0.26	1.1 [0.7–1.7]	0.63
Prostate	425	114	26.8	1.4 [0.9–2.0]	0.09	1.0 [0.7–1.4]	0.93
Lung	412	144	35.0	2.0 [1.4–2.9]	<0.01	2.4 [1.7–3.5]	<0.01
Urinary tract	235	68	28.9	1.5 [1.0–2.3]	0.04	1.3 [0.9–2.0]	0.20
Female genital organs	66	13	19.7	0.9 [0.5–1.8]	0.79	1.4 [0.7–2.9]	0.32
Lip, oral cavity and pharynx	117	26	22.2	1.1 [0.6–1.8]	0.82	1.5 [0.9–2.6]	0.12
Skin	78	20	25.6	1.3 [0.7–2.3]	0.40	1.0 [0.5–1.8]	0.91
Other cancers **	249	67	26.9	1.4 [0.9–2.1]	0.13	2.0 [1.3–3.0]	<0.01

OR: odds ratio; aOR: adjusted odds ratio. * adjusted on age, sex, dementia, heart failure, chronic respiratory disease, cirrhosis, diabetes, deficiency anemia and pulmonary bacterial infection. ** including mesothelial and soft issue, respiratory and intrathoracic organs (except lung), bone and articular cartilage, endocrine glands, other male cancers and central nervous system.

## Data Availability

Data sharing not applicable. Indeed, the PMSI database was transmitted by the national agency for the management of hospitalization data. and we are not allowed to transmit these data. PMSI data are available for researchers who meet the criteria for access to these French confidential data (this access is submitted to the approval of the National Committee for data protection) from the national agency for the management of hospitalization (ATIH—Agence technique de l’information sur l’hospitalisation). Address: Agence technique de l’information sur l’hospitalisation, 117 boulevard Marius Vivier Merle, 69329 Lyon Cedex 03.

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
