# Peer review of "Comparison of Cancer Patients to Non-Cancer Patients among COVID-19 Inpatients at a National Level"

_cancers, 2021, doi:10.3390/cancers13061436_

Round 1
Reviewer 1 Report
We suggest to underline in the discussion that Covid-19 infection involve the immunological system just from its offset, and that cancer patients are often immunocompromised or immunosupressed just for their basic disease or subsequent treatment. Secondly, in its evolution Covid - 19 can become a proper immuno-inflammatory disease, which impact on its outcome. ( L. Roncati et al. Clin Immunol 2020; 217: 108487)
Author Response
R1
Review Report Form
We thank the Reviewer for having given us an opportunity to substantially improve the content and the presentation of our manuscript. We have modified the article accordingly to your requests. You will find every modification in the text using track changes, and the pages are noted in the answer for every point below. We hope we have met your requirements to improve this paper.
Open Review
(x) I would not like to sign my review report ( ) I would like to sign my review report English language and style ( ) Extensive editing of English language and style required ( ) Moderate English changes required
(x) English language and style are fine/minor spell check required ( ) I don't feel qualified to judge about the English language and style
Yes Can be improved Must be improved Not applicable
Does the introduction provide sufficient background and include all relevant references? (x) ( ) ( ) ( )
Is the research design appropriate? (x) ( ) ( ) ( )
Are the methods adequately described? (x) ( ) ( ) ( )
Are the results clearly presented? (x) ( ) ( ) ( )
Are the conclusions supported by the results? (x) ( ) ( ) ( )
Comments and Suggestions for Authors
We suggest to underline in the discussion that Covid-19 infection involve the immunological system just from its offset, and that cancer patients are often immunocompromised or immunosupressed just for their basic disease or subsequent treatment. Secondly, in its evolution Covid - 19 can become a proper immuno-inflammatory disease, which impact on its outcome. ( L. Roncati et al. Clin Immunol 2020; 217: 108487)
We thank the reviewer for this helpful comment. We have now added a few sentences and references on the potential underlying mechanisms associated with the higher mortality we observed, as follows (Discussion, Page 14):
“The mechanisms underlying the progression of COVID-19 to a severe form may not all be the same since not all patients with cancer are at a similarly high risk of developing severe COVID-19. These mechanisms certainly include not only host related factors such as age or cachexia, cancer- or drug-induced hypercoagulability states, or the possible hyperexpression of entry factors such as angiotensin-converting enzyme 2 or neurophilin-1 (Cantuti-Castelvetri L et al Neuropilin-1 facilitates SARSCoV-2 cell entry and provides a possible pathway into the central nervous system. bioRxiv 2020) but also cancer-related myeloid cell dysfunction and T-cell exhaustion, both of which impair antiviral immunity (McKinney EF, Smith KGC. Metabolic exhaustion in infection, cancer and autoimmunity.Nat Immunol 2018; 19:213–21). On the other hand, the immunological system is also involved in the subsequent virus-driven cytokine storm (Roncati et al. Type 3 hypersensensitivity in COVID-19 vasculitis. Imm Clin 2020), this cancer- or drug- associated immune-depression may temper or delay it (Moore JB, June CH. Cytokine release syndrome in severe COVID-19. Science 2020; 368:473–4).”

Reviewer 2 Report
Comparison of cancer patients to non-cancer patients among COVID-19 inpatients at a national level
Summary: In this article, the authors have compared the rate of mortality of cancer patients in France that are hospitalized after COVID19 infection. This study shows that COVID19 patients with cancer have two-fold higher risk of death after hospitalization. The authors have done commendable job of collecting and analyzing data across all the hospital across France to show an increase in mortality rate of cancer patients with COVID19 infection. However, this article suffers from lack of originality as many authors already have suggested that COVID19 infection might increase mortality (Kuderer, N. M., et al. Lancet 2020). Secondly, all the data in this study is correlative and therefore does not add any impact for publishing this study in Cancers journal. With this I would regret to inform to authors that I have very diminished enthusiasm to accept this article for publication.
Author Response
R2
Review Report Form
We thank the Reviewer for having given us an opportunity to substantially improve the content and the presentation of our manuscript. We have modified the article accordingly to your requests. You will find every modification in the text using track changes, and the pages are noted in the answer for every point below. We hope we have met your requirements to improve this paper.
Open Review
(x) I would not like to sign my review report ( ) I would like to sign my review report English language and style ( ) Extensive editing of English language and style required
(x) Moderate English changes required
( ) English language and style are fine/minor spell check required ( ) I don't feel qualified to judge about the English language and style
The manuscript has been entirely reviewed by a native English speaker specialized in proof-reading articles.
Yes Can be improved Must be improved Not applicable
Does the introduction provide sufficient background and include all relevant references? (x) ( ) ( ) ( )
Is the research design appropriate? ( ) (x) ( ) ( )
Are the methods adequately described? ( ) (x) ( ) ( )
Are the results clearly presented? ( ) (x) ( ) ( )
Are the conclusions supported by the results? ( ) (x) ( ) ( )
Comments and Suggestions for Authors
Comparison of cancer patients to non-cancer patients among COVID-19 inpatients at a national level
Summary: In this article, the authors have compared the rate of mortality of cancer patients in France that are hospitalized after COVID19 infection. This study shows that COVID19 patients with cancer have two-fold higher risk of death after hospitalization. The authors have done commendable job of collecting and analyzing data across all the hospital across France to show an increase in mortality rate of cancer patients with COVID19 infection. However, this article suffers from lack of originality as many authors already have suggested that COVID19 infection might increase mortality (Kuderer, N. M., et al. Lancet 2020). Secondly, all the data in this study is correlative and therefore does not add any impact for publishing this study in Cancers journal. With this I would regret to inform to authors that I have very diminished enthusiasm to accept this article for publication.
We agree with the Reviewer that the contribution of our study was not sufficiently highlighted.
This topic was investigated in some papers early in the epidemic, seeing as it was important to identify cancer patients at risk of severe Covid-19 outcomes. Published studies include Kuderer's article (Lancet, June 2020) which was the first to report on a large population of cancer patients (nearly 1,000 patients). However, this study called for further investigations.
First, the outcome on mortality was limited to 30 days.
Second, as stated by the authors, some notable regional variations in the outcomes exist: “the Spanish subgroup had no ICU admissions and no patients put on mechanical ventilation, but had ten deaths. The Canadian subgroup had the highest proportion of patients admitted to hospital, yet had the numerically lowest rate of deaths of any of the regional subgroups. These findings, including the reduced risk of 30-day all-cause mortality associated with residence in Canada and the US-Midwest probably reflect regional differences in the response to COVID-19, and different timelines of the local pandemic, and deserve further study.”
Finally, as also noted by the authors, there were “not able to do an analysis comparing our cohort [cancer patients with COVID-19] with patients [….] without cancer with COVID-19. Such an analysis would better place the current data into a larger context.”
Our study thus provides additional information to Kuderer's article as the objective of our study was to compare patients with cancer and patients without cancer, thanks to our COVID-19 cohort. The first advantage of this study is that it is based on a national cohort including all patients hospitalized in France, and it is therefore representative of the population of an entire country. The second advantage is that it includes a very large number of patients, nearly 90.000 COVID-19 patients, including 6.000 cancer patients. The third advantage is the relatively homogeneous level of care across the country and therefore across the entire cohort. The last advantage is that we were able to study mortality up to the end of the hospital stay without limiting ourselves to 30 days. The maximum time to in-hospital death observed in our study ranged between 61 days (hematological cancers) and 95 days (no cancer). We added this point in the Results section (page 11).
This is therefore, to our knowledge, the largest study in the world so far, and, moreover, it is a representative study that includes all the data for a whole country. Above all, this study answers a question that was not addressed in Kuderer's article and which was highlighted by the authors as a fundamental issue for clinical management and the organization of preventive measures: Cancer kills and COVID kills, but do patients with a cancer diagnosis die more than other COVID patients? This question has not been addressed on a large scale elsewhere.
We highlighted the contribution of our study in the Discussion, page 18: “To our knowledge, this is the largest study in the world so far, and the only one to provide comprehensive data for an entire country. Our study compares cancer patients with COVID-19 to non-cancer patients with COVID-19. This answers a question that was highlighted by Kuderer et al in their pioneering work (Kuderer et al Lancet 2020) as a fundamental issue for clinical management and the organization of preventive measures, and that is not addressed on a large scale elsewhere.”
Reviewer 3 Report
This manuscript offers summary statistics about COVID-19 fatality in hospital in the midst of COVID-19 pandemic. Although the study samples are limited to particular patient groups, it is still useful for the planning of hospital resource distribution, which is critical for COVID-19 patient caring.
However, I am worried about some biases that did not seem to be accounted for.
As hospitalisation criteria were different between the Stages, hospital patients should be stratified by the Stage too.
Patients could be grouped by hospital type: private or public. It may also be associated with social deprivation score, but could be a non-negligible factor.
Below are some additional comments.
Some stats in Table 3 have large confidence intervals. They should be discussed.
For patients with older age-groups (>80), the percentage of ICU support is low, but their in-hospital mortality is high. This could be an interesting discussion point, but it was hardly discussed.
In lines 279-281 (‘Covid19 affects the most underprivileged. These results lead us to believe that COVID-19 affects disadvantaged populations even more than cancer.’), it isn’t clear on what basis this claim was made. In my understanding of this manuscript, social deprivation score was compared only for those with cancer to those without cancer, and those with different cancer types.
Also regarding social deprivation score, it would be more useful to have the quartiles (lower, medium and upper) in relevant tables rather than the number of patients in different social deprivation score groups. They should be evenly distributed between these groups by the definition of quartiles.
Much of the ‘Discussion’ part is repeat of results. It could be reorganised.
Figure 1 needs sub-labelling, such as Figure 1A, 1B, etc.
Author Response
R3
Review Report Form
We thank the Reviewer for having given us an opportunity to substantially improve the content and the presentation of our manuscript. We have modified the article accordingly to your requests. You will find every modification in the text using track changes, and the pages are noted in the answer for every point below. We hope we have met your requirements to improve this paper.
Open Review
(x) I would not like to sign my review report ( ) I would like to sign my review report English language and style ( ) Extensive editing of English language and style required ( ) Moderate English changes required
(x) English language and style are fine/minor spell check required ( ) I don't feel qualified to judge about the English language and style
Yes Can be improved Must be improved Not applicable
Does the introduction provide sufficient background and include all relevant references? (x) ( ) ( ) ( )
Is the research design appropriate? ( ) ( ) (x) ( )
Are the methods adequately described? ( ) (x) ( ) ( )
Are the results clearly presented? ( ) (x) ( ) ( )
Are the conclusions supported by the results? ( ) ( ) (x) ( )
Comments and Suggestions for Authors
This manuscript offers summary statistics about COVID-19 fatality in hospital in the midst of COVID-19 pandemic. Although the study samples are limited to particular patient groups, it is still useful for the planning of hospital resource distribution, which is critical for COVID-19 patient caring.
However, I am worried about some biases that did not seem to be accounted for.
As hospitalisation criteria were different between the Stages, hospital patients should be stratified by the Stage too.
We agree with this remark, even though most COVID-19 patients were hospitalized during the Stage 3 period (95%). We performed an analysis limited to the Stage 3 COVID-19 pandemic period as a sensitivity analysis, which showed similar results to that of the global analysis. We added the following sentence at the end of the Results section (page 12):
“Our second series of sensitivity analysis, which was restricted to Stage 3 of the epidemic, which represents 95% of the cohort (March 14th onwards), provided similar results (Supplementary Tables 8 to 12)
This was added in the “Discussion” section (page 17):
“Finally, we conducted several sensitivity analyses that allowed us to consolidate our results. In particular, we restricted the period to the stage 3 of the epidemic, showing similar results to those of the main analysis.”
We also mentioned this sensitivity analysis in the “Methods” section (page 9):
“Considering the potential heterogeneity of patients admitted to hospital in the three stages of the epidemic, we finally performed a second series of sensitivity analyses restricted to Stage 3 of the epidemic (March 14th onwards). ”
Patients could be grouped by hospital type: private or public.
We agree with the Reviewer’s suggestion. We first described hospital type, private or public, among admitted patients (see Table 2). We added this variable in the Methods section (page 7) and found that public hospitals admitted 89.1% of patients without cancer, 92.7% of patients with hematological cancer, 84.1% of patients with solid metastatic cancer, and 88.3% of patients with solid cancer without metastasis (Results, page 9).
We then implemented 2-level hierarchical models to study the risk of death. We used the individual variables as 1st level and the variables related to the hospitals as the 2nd level. The results were similar to the main analysis.
We have described this modeling as a sensitivity analysis in the Methods section (page 8) and have added in the Results section that the results were identical to the main analysis (page 12). We also provided the adjusted odds ratios in Supplementary Tables 6 and 7.
It may also be associated with social deprivation score, but could be a non-negligible factor.
We also implemented 2-level hierarchical models to account for this socio-demographic factor on the risk of death. We used the individual variables as 1st level and the social deprivation score (geographical unit) as the 2nd level. The results were also similar to the main analysis.
We have described this modeling as a sensitivity analysis in the method section (page 8) and have added in the result section that the results were identical to the main analysis (page 12). We also provided the adjusted odds ratios in Supplementary Tables 6 and 7.
Below are some additional comments.
Some stats in Table 3 have large confidence intervals. They should be discussed.
We agree that due to smaller group sizes, the confidence intervals are large for the younger patients (less than 40 years). We added this point in the Discussion section (page 16):
“However, even if the national database made it possible to refine the analysis in terms of age classes, due to the smaller group sizes, the confidence intervals of adjusted odds ratios are large for patients younger than 40 years.”
For patients with older age-groups (>80), the percentage of ICU support is low, but their in-hospital mortality is high. This could be an interesting discussion point, but it was hardly discussed.
We agree that this is an interesting point to be discussed. We added the following sentences in the Discussion (Page 16):
“On the other hand, cancer patients in the oldest age group (>80 years) had a slightly lower percentage of ICU support, although non-significant, which contrasts with their high in-hospital mortality. This could reflect the fact that patients may have been assessed for ICU access in terms of clinical status and prognosis.”
In lines 279-281 (‘Covid19 affects the most underprivileged. These results lead us to believe that COVID-19 affects disadvantaged populations even more than cancer.’), it isn’t clear on what basis this claim was made. In my understanding of this manuscript, social deprivation score was compared only for those with cancer to those without cancer, and those with different cancer types.
We fully agree that there was an insufficient basis for this claim. We modified the paragraph accordingly (Discussion section, page 13):
“As regards social deprivation, cancer is more frequent in low-income populations in France, with the exception of breast cancer [21]. COVID-19 also affects the most underprivileged. However, our data indicates that COVID-19 patients without cancer were more underprivileged than COVID-19 patients with cancer. Further studies are needed to better understand the relationship between COVID-19, cancer, and deprivation..”
Also regarding social deprivation score, it would be more useful to have the quartiles (lower, medium and upper) in relevant tables rather than the number of patients in different social deprivation score groups. They should be evenly distributed between these groups by the definition of quartiles.
We estimated quartiles of the social deprivation score defined on the overall population of 89,540 COVID-19 patients to compare the distribution between groups. We clarified this point in the Methods section (page 7) and specified the values of the classes in Table 2.
Here are also the values of the quartiles for the different groups:
|
|
Lowest |
Second |
Third |
Highest |
|
Solid metastatic cancer |
< -1.377 |
[-1.377;-0.205[ |
[-0.205;0.851[ |
≥ 0.851 |
|
Solid cancer without metastasis |
< -1.393 |
[-1.393;-0.210[ |
[-0.210;0.773[ |
≥ 0.773 |
|
Hematological cancer |
< -1.405 |
[-1.405;-0.171[ |
[-0.171;0.851[ |
≥ 0.851 |
|
Without cancer |
< -1.219 |
[-1.219;-0.143[ |
[-0.143;0.928[ |
≥ 0.928 |
Much of the ‘Discussion’ part is repeat of results. It could be reorganised.
The discussion has been extended beyond the results and we tried to avoid repeating when possible. It has also been reorganized
Figure 1 needs sub-labelling, such as Figure 1A, 1B, etc.
Thank you for this suggestion. Sub-labelling has been provided.
Round 2
Reviewer 3 Report
The authors seem to have limited the patients to a certain stage or to a certain group of hospitals for the sensitivity analysis, even though it was stated that the patients were stratified in the sensitivity analysis. Maybe it wasn’t very clearly presented in the manuscript. However, if the sensitivity analyses do not contradict the main results, it should be a minor issue to rectify.
Just to add, the analysis with SDI, maybe it is more intuitive to show the proportion of patients in different groups by the quartiles of national SDI, instead of showing quartiles of each group.
Author Response
Response to Reviewer 3
Comments and Suggestions for Authors
The authors seem to have limited the patients to a certain stage or to a certain group of hospitals for the sensitivity analysis, even though it was stated that the patients were stratified in the sensitivity analysis. Maybe it wasn’t very clearly presented in the manuscript. However, if the sensitivity analyses do not contradict the main results, it should be a minor issue to rectify.
We thank the Reviewer for this comment. We agree that it was not clearly stated in the paper. The sensitivity analyses relative to stage were restricted to the period March 14th – onward, while the sensitivity analyses relative to the type of hospital were not restricted and took into account all hospitals (public and private). We clarified this point in the Methods section (page 8):
“We also performed sensitivity analyses using 2-level hierarchical models. The first model used the individual variables as the 1st level, and the type of hospital (public or private) as the 2nd level.”
Just to add, the analysis with SDI, maybe it is more intuitive to show the proportion of patients in different groups by the quartiles of national SDI, instead of showing quartiles of each group.
We agree that it would be an interesting alternative to present the analyses with SDI according to the quartiles of its national distribution. However, we did not have access to these values. In the paper, we provided the quartiles of our overall study population whatever the cancer presence and cancer type group (Table 2), and conducted the analyses according to these categories.